


# A Novel Global Freeze-Thaw State Detection Algorithm Based on Passive L-Band Microwave Remote Sensing

Shaoning Lv[1,2*], Clemens Simmer[2,5], Yijian Zeng[3], Jun Wen[4], Yuanyuan Guo[1], Zhongbo Su[3]

[1]Department of Atmospheric and Oceanic Sciences & Institute of Atmospheric Sciences, Fudan University, 200438, Shanghai, China
[2]Institute for Geosciences - Meteorology at the University of Bonn, Auf dem Huegel 20, 53121 Bonn, Germany
[3]Department of Water Resources, Faculty of Geo-information Science and Earth Observation (ITC), University of Twente, P.O. Box 217, 7500AE, Enschede, The Netherlands
[4]the Plateau Atmosphere and Environment Key Laboratory of Sichuan Province, Chengdu University of Information Technology, Chengdu 610225, Sichuan, China
[5]Cloud and Precipitation Exploration Laboratory (CPEX-Lab) of Geoverbund ABC/J, Auf dem Huegel 20, 53121 Bonn, Germany

*Correspondence to*: Shaoning Lv (lvshaoning@fudan.edu.cn)

**Abstract.** Knowing the Freeze-Thaw (FT) state of the land surface is essential for many aspects of weather forecasting, climate, hydrology, and agriculture. Near-surface air temperature and land surface temperature are usually used in meteorology to infer the FT-state. However, the uncertainty is large because both temperatures can hardly be distinguished from remote sensing. Microwave L-band emission contains rather direct information about the FT-state because of its impact on the soil dielectric constant, which determines microwave emissivity and the optical depth profile. However, current L band-based FT algorithms need reference values to distinguish between frozen and thawed soil, which are often not known sufficiently well.

We present a new FT-state detection algorithm based on the daily variation of the H-polarized brightness temperature of the SMAP L3c FT global product for the northern hemisphere, which is available from 2015 to 2021. The exploitation of the daily variation signal allows for a more reliable state detection, particularly during the transitions periods, when the near-surface soil layer may freeze and thaw on sub-daily time scales. The new algorithm requires no reference values; its results agree with the SMAP FT state product by up to 98% in summer and up to 75% in winter. Compared to the FT state inferred indirectly from the 2-m air temperature of the ERA5-land reanalysis, the new FT algorithm has a similar performance as the SMAP FT product. The most significant differences occur over the midlatitudes, including the Tibetan plateau and its downstream area. Here, daytime surface heating may lead to daily FT transitions, which are not considered by the SMAP FT state product but are correctly identified by the new algorithm. The new FT algorithm suggests a 15 days earlier start of the frozen-soil period than the ERA5-land's 2-m air temperature estimate. This study is expected to extend L-band microwave remote sensing data for improved FT detection.

Keywords: frozen soil state estimation, passive microwaves, remote sensing, SMAP FT state product

## 1. Introduction

Spatial patterns and the timing of freeze-thaw (FT) state transitions over land are highly variable; they strongly impact land-atmosphere interactions and thus weather, climate and hydrological, ecological, and biogeochemical processes (Walvoord and Kurylyk, 2016; Schuur et al., 2015; Zeng et al., 2019; Yu et al., 2020b). In particular, FT state transition leads to differences in hydrological and thermal conductivities/diffusivities, the albedo for solar and emissivity for terrestrial radiation, and latent/sensible heat fluxes (Hu et al., 2017; Gao et al., 2016; Zhao et al., 2017). The albedo is, e.g., higher for frozen than for unfrozen soil, and the water and energy exchange between the land surface and the atmosphere is reduced while frozen because





of a much weaker surface heating due to melting. Changes in the FT state dynamics can also signal climate change (Schuur et al., 2015) and invoke permafrost carbon feedback (Zhao et al., 2018a). Besides, ecosystem responses to seasonal FT-state changes are rapid via significant changes in evapotranspiration, soil respiration, plant photosynthetic activity, liquid water availability, vegetation net primary production, and net ecosystem $CO_2$ exchange (NEE) with the atmosphere(Kimball et al., 1997; Li et al., 2014; Cramer et al., 1999; Matzner and Borken, 2008; Wang et al., 2016). Thus, the knowledge of the FT state

is required for modeling work in the above subjects, which invoke different parametrizations for frozen and unfrozen soil (Xie et al., 2018; Swenson et al., 2012; Yu et al., 2020a; Yu et al., 2018; Mwangi et al., 2020).

However, FT state estimations from in-situ temperature observations are limited in scale, and it is not straightforward to deduce the state from soil, skin, or near-surface air temperature. In contrast, more direct state information results from the very different microwave dielectric constant for frozen and unfrozen soil (Yashchenko and Bobrov, 2016; Schwank et al., 2004;

Rautiainen et al., 2014). The difference leads to emissivities of ~0.6 for unfrozen and ~0.9 for frozen soil, with a much deeper penetration depth for the latter (Lv et al., 2019; Zhao et al., 2018b; Lv et al., 2018). Accordingly, microwave brightness temperatures (*TBs*) change sharply during FT state transitions. For instance, NASA's MEaSUREs (Making Earth System Data Records for Use in Research Environments) program provides two global daily products for the land FT state based on a single-channel algorithm (Kim et al., 2017b). One covers the years 1979 to 2017 and exploits the 37 GHz channels of three

satellite-based passive microwave sensors: the Scanning Multichannel Microwave Radiometer (SMMR), the Special Sensor Microwave/Imager (SSM/I), and the Special Sensor Microwave Imager/Sounder (SSMIS). For MEaSUREs, 37 GHz is selected because of its high correlation with the near-surface air temperature. Moreover, lower-frequency L-band sensors on SMOS (Soil Moisture and Sea Salinity) and SMAP (Soil Moisture Active and Passive) are more suitable for FT-change detection because of their deeper penetration depth (Xu et al., 2018; Rautiainen et al., 2016). At the L-band, the cross-polarized

gradient ratio (XPGR) between H and V polarization is used to analyze the SMAP observations. Similar to the higher-frequency single-channel algorithms, the use of XPGR needs reference values for thawed and frozen soil states and a threshold value for the discrimination between both (Rautiainen et al., 2014; Kim et al., 2017a). The current SMAP FT state detection requires at least 20 days for finding reference values for the frozen state, which can be challenging for short interim frozen periods induced, e.g., by synoptic-scale cold waves.

Unlike the XPGR method or single-channel algorithms, we derive a new FT algorithm based on sub-daily *TB* changes, which overcomes the limitations in identifying the frozen/thawed *TB* reference values. The new algorithm hypothesizes that FT state changes are dynamic and complex and vary continuously in space and time. The hypothesis is functional because the conditions driving FT changes, e.g., radiation balance and air temperature, change on broad time scales, reaching from sub-daily, daily, synoptic, seasonal, annually, to interdecadal (Guo and Wang, 2014). In particular, daily FT cycles signal the

beginning and endings of the annual FT cycle. Especially in cold arid regions, which are prone to experience FT state transitions, soil moisture fluctuations due to evaporation and precipitation, and their L-band signals are comparatively low on daily and synoptic scales. The potentially large daily surface soil temperature variations only weakly influence the L-band microwave emitting temperature because of the large L-band penetration depths, which give more weight to deeper and thus more constant soil temperatures.

In this study, we use the daily *TB* cycle and its connection with changing penetration depths during FT state changes to develop a new FT state detection algorithm, which exploits the daily *TB* differences ($\left|\Delta TB\right|$) between 6 am and 6 pm (Local Time) - the overpassing times for the SMAP and SMOS satellites. The method is based on microwave transfer theory and does not need reference values. The structure of the paper is as follows: Section 2.1 describes the data used, including the SMAP FT product, the ERA5-land reanalysis. Section 2.2 details the SMAP FT product followed by our new method in Section 2.3.

The statistics required for implementing the new method are explained in Section 2.4. Results are found in Section 3.1. Section 3.2 compares the new FT algorithm with the current SMAP FT product, and section 3.3 quantifies the uncertainties of the new



method. Conclusions and a discussion on the relation between 2-m air temperature and soil FT states are presented in Section 4.

## 2. Methodology

### 2.1 Data and its illustration at the Xilinhot site

We use the following three data sets in this study:

1) SMAP *TB* observations and the derived FT-state indicator (Xu et al., 2018) with 36 km x 36 km spatial resolution at 6 am and 6 pm local time. SMAP's L3 product includes an FT state indicator besides H and V polarized TB observations at L-band (1.41 GHz). The data is available starting March 30, 2015 (Xu et al., 2018). The derived FT-state indicator (SMAP L3c FT product) is available globally from 85.044°S to 85.044°N. We use the SMAP *TB*s for the new algorithm and the binary FT-state indicator for frozen (1) and thawed soil (0), including the transition direction for its evaluation.

2) Hourly 2m-air ($T_{2m}$), skin ($T_{skin}$), and soil temperatures from the ERA5-Land reanalysis (Muñoz-Sabater et al., 2021). ERA5-Land provides a consistent representation of the evolution of land state variables over several decades at a higher resolution (0.1° x 0.1°) than ERA5 (0.25° x 0.25°). ERA5-land has been produced by replaying the land component of the ECMWF ERA5 climate reanalysis at an enhanced resolution. ERA5-Land also provides soil profile information that is vital to the analysis of L-band *TBs*. Since L-band observations may have – depending on the FT-state - deep penetration depths, neither particular variable such as 2m-air, skin, and soil temperatures with diurnal changes in ERA5-land is suitable for comparison with the daily FT indicator at a daily scale. Besides, the daily 2m-air temperature ($T_{2m}$) is used as an FT state reference to evaluate the existing and the new FT algorithms. Considering the detectable range of the L-band, FT state reference can be inferred from 2-m air/skin/5 cm soil/10cm soil temperatures. However, the inferred FT state from these temperatures may contradict each other at the moment, and it is hard to judge which one represents the signal detected by the L-band. In SMAP FT Cal/Val, the in-situ 2m-air temperature or the soil temperature at 5cm is used to validate and calibrate SMAP's FT state indicator(Dunbar et al., 2020). The in-situ soil moisture at 10 cm and the skin temperature are taken as FT state reference, not as ground truth in the SMAP's FT Cal/Val. $T_{2m}$ is often used to estimate soil FT states (Baker and Ruschy, 1995; Fortin, 2010; Ho and Gough, 2006). . It shall be noted that $T_{2m}$ is not the condition for judging frozen/thawed soil from the reanalysis but an indicator for the thermal conditions near the surface regarding the land-atmosphere interaction.

To better illustrate how the new FT algorithm works, we selected the location of the Xilinhot site already used in other microwave remote sensing studies (Yamano et al., 2003; Shi et al., 2011; Wu and Chen, 2012) to illustrate the functioning of the new FT detection algorithm because of its meteorological conditions, which are typical for regions experiencing complex FT-state changes. Xilinhot is located in the mid-latitude semi-arid grassland in Inner Mongolia, one of the main surface types on the globe; 31-43% of the land cover in the northern hemisphere belongs to this climate type (Chapin et al., 2013). From the ERA5-land reanalysis data, we use the nearest to Xilinhot to represent the site's land surface and meteorological conditions. From the data for 6 am and 6 pm local time (UTC+08), the daily differences of $T_{2m}$, $T_{skin}$, and soil temperature at 0-7 cm ( *stl*1 ), 7-28 cm ( *stl*2 ), 28-100 cm ( *stl*3 ), and 100- 189 cm ( *stl*4 ) are computed and used for interpreting the satellite-observed *TB* signals for the location of the Xilinhot site.

Figure 1: SMAP *TB* time series and the derived reference FT state (grey for frozen and white for unfrozen) extracted for the location of the Xilinhot site.

### 2.2 the SMAP F/T algorithm





The SMAP FT algorithm is based on the so-called relative frost factor $FF_{rel}$,

$$FF_{rel} = \frac{FF_{NPR} - FF_{fr}}{FF_{th} - FF_{fr}} \qquad (1)$$

120  where $FF_{NPR}$ is the frost factor defined as the normalized polarization ratio,

$$FF_{NPR} = \frac{TB_v - TB_h}{TB_v + TB_h} \qquad (2)$$

and $FF_{fr}/FF_{th}$ is the reference frost factor for the frozen/thawed state, respectively. $FF_{fr}$ is the average $FF_{NPR}$ for January and February (winter), and $FF_{th}$ is the average $FF_{NPR}$ of July and August (summer) over the years 2015–2018.

The SMAP FT status ($FT_{SMAP}$) is derived from $FF_{rel}$ for each location via

125
$$FT_{SMAP} = \begin{cases} thaw, if\, FF_{rel} > threshold \\ frozen, if\, FF_{rel} \leq threshold \end{cases} \qquad (3)$$

where a threshold of 0.5 was used globally.

The algorithm relies on the quality of the two reference values $FF_{fr}$ and $FF_{th}$. Their estimation requires at least 20 days of relatively stable frozen (or unfrozen) conditions (Derksen et al., 2017). $FF_{th}$ is hard to be identified at higher latitudes and altitudes where the ground is frozen throughout the year, while $FF_{fr}$ is hard to determine for the midlatitudes where the soil is 130  not completely frozen from the surface down to the L-band penetration depth. According to the SMAP FT handbook (Dunbar et al., 2020), $FF_{NPR}$ needs to be larger than an arbitrary value of 0.1, which excludes relatively dry areas that undergo minor dielectric constant changes during FT transitions. As an extended algorithm, FT-SCV(Freeze/Thaw algorithm using Single Channel TBV) is used to overcome this defect in $FF_{NPR}$. FT-SCV does not reply on the freeze/thaw reference derived from the winter/summer period, but its correlation with surface air temperature from global weather stations and reanalysis data(Dunbar 135  et al., 2020).

**2.3 The new FT algorithm**

The new FT algorithm uses the strong $TB$ variations over the day caused by freezing in the night and thawing over the day, which happens over a period of days at the beginning and end of the totally frozen period. The new FT algorithm has a similar basis to the Diurnal Amplitude Variation (DAV) approach applied to higher frequency passive microwave measurements for 140  snow and ice sheet applications(Kopczynski et al., 2008; Tedesco, 2007). For L-band, which is longer than previous DAV applications, the signal can penetrate ice and snow over the soil surface and be related to the FT state of the soil. To retrieve this signal from the microwave transfer theory, we start with the zeroth-order microwave transfer model given by

$$TB = \varepsilon T_{eff} \qquad (4)$$

where $\varepsilon$ is the emissivity, which depends on the soil dielectric constant and mainly varies with soil moisture and the FT-state 145  of the soil. $T_{eff}$ is the vertically integrated soil temperature profile weighted with the soil dielectric profile as (Lv et al. 2016a)

$$T_{eff} = T_1\left(1 - e^{-\tau_1}\right) + \sum_{i=2}^{n-1} T_i\left(1 - e^{-\tau_i}\right)\prod_{j=1}^{i-1} e^{-\tau_j} + T_n \prod_{j=1}^{n-1} e^{-\tau_j} \qquad (5)$$





where $T$ is soil temperature, $\tau$ is soil optical depth, and the subscripts $i$ and $j$ are the layer numbers counting from the top of the soil (1) to the bottom of a layered soil slab ($n$) influencing $TB$. Because of the much deeper penetration depth of frozen soil, the attenuation of radiation emitted from lower layers is strongly reduced (Lv et al., 2016), which enlarges the depth down to
which the integration for $T_{eff}$ must be performed; thus, the deeper soil layers with their only minor daily and even seasonally varying temperatures dominate $TB$s of frozen soil (see $TB$ variations during winter in Fig. 1). Thus, especially in winter $TB$s of frozen soil are mostly higher than of unfrozen soil.

When unfrozen, soil moisture variations due to evaporation lead to $TB$ increases of only up to 10 K during a day. An exception of significant daily $TB$ changes for unfrozen soil is precipitation, which can reduce $TB$s by tens of K. $TB$ can change
in the same range due to daily soil temperature variations via $T_{eff}$ ((Eqs. (4) & (5)). $TB$-changes during FT-transitions are in the range and larger than the precipitation signal because of the huge $\varepsilon$ difference between frozen and unfrozen soil. When frozen, emissivity – and thus $TB$ - variations - are very small and only slightly depend on soil composition, such as the clay/sand fraction and organic matter, which also affect the emissivity of unfrozen soil. Thus, daily $TB$ changes for unfrozen soil – except for precipitation – are much smaller than those caused by freezing and thawing. Any FT transition typically begins and ends
at the surface, thus, L-band radiometers can sense the start and end of FT transitions. The new FT algorithm exploits the daily $TB$ difference caused by FT state transitions and puts the transitions between the annual frozen and unfrozen soil periods in the middle of the detected transition period exhibiting daily FT-state cycles.

We use the following formalism for the FT-state detection. Let

$$\begin{cases} TB_{ih\_6am} = \varepsilon_{6am} T_{eff\_6am} \\ TB_{ih\_6pm} = \varepsilon_{6pm} T_{eff\_6pm} \end{cases} \tag{6}$$

$TB_{ih\_6am} / TB_{ih\_6pm}$ are the $TB$s observed by SMAP at 6 am and 6 pm local time on day $i$ in H-polarization (h) with $\varepsilon_{6am} / \varepsilon_{6pm}$ the respective soil emissivities and $T_{eff\_6am/pm}$ the respective $T_{eff}$. When the soil surface is frozen both at 6 am and 6 pm, and neglecting the impact of soil temperature changes on the dielectric constant, i.e., $\varepsilon_{6am} = \varepsilon_{6pm} = \varepsilon$, the $TB$ difference between both is using Eq. (6) given by

$$\begin{aligned} \Delta TB_i &= TB_{ih\_6pm} - TB_{ih\_6am} \\ &= \varepsilon \left( T_{eff\_6pm} - T_{eff\_6am} \right) \\ &= \varepsilon \left[ \Delta T_1 \left( 1 - e^{-\tau_1} \right) + \sum_{i=2}^{n-1} \Delta T_i \left( 1 - e^{-\tau_i} \right) \prod_{j=1}^{i-1} e^{-\tau_j} + \Delta T_n \prod_{j=1}^{n-1} e^{-\tau_j} \right] \end{aligned} \tag{7}$$

At 6 am/pm, soil temperature and moisture profile gradients are less sharp than at noon, and $\Delta TB_i$ will be much smaller than the temperature differences ($\Delta T_i$) in any layer, since $\varepsilon < 1$, $\left( 1 - e^{-\tau_1} \right) < 1$, $\left( 1 - e^{-\tau_i} \right) \prod_{j=1}^{i-1} e^{-\tau_j} < 1$ and $\prod_{j=1}^{n-1} e^{-\tau_j} < 1$, i.e.,

$$\left| \Delta TB_i \right| < \max(\left| \Delta T_i \right|) \tag{8}$$

Eq. (8) is not valid for unfrozen soil because $\varepsilon$ will change with soil moisture over the day due to evaporation and precipitation, which will dominate $\Delta TB_i$. But $\Delta TB_i$ will also be small when no precipitation happens between both times and when the sky
is cloudy, and low winds reduce evaporation. Thus, $\Delta TB_i$ is not enough to infer the FT state. A sudden heat/cold wave can interrupt a daily FT state transition, which may induce a large $\Delta TB_i$ with soil staying frozen or unfrozen throughout the event.





Such synoptical scale heat/cold waves make it difficult to identify the beginning/end of the yearly freezing. To avoid this problem, we define the beginning/end of the annual freezing as the first/last day of the frozen soil in an annual freezing cycle and ignore the interruption by such heat or cold waves.

Thus, to filter out the influence of synoptic variations and cloudy and/or low wind days, we use in addition the $\Delta TB_i$ variance over $\beta$ days

$$\text{var}(\Delta TB)_\beta = \frac{1}{\beta} \sum_{i=-(\beta-1)/2}^{i=(\beta-1)/2} \left[ \Delta TB_i - E\left( \Delta TB_i \right) \right]^2 \tag{9}$$

$\text{var}(\Delta TB)_\beta$ is not a new parameter but to keep $\left| \Delta TB_i \right|$ filtering out the synoptic weather interference. The selection $\beta=7$ optimally filters out the impact of atmospheric Rossby waves in the midlatitudes (3-5 days) at locations experiencing annual

FT cycling in the mid-latitudes (Blackmon, 1976). This averaging will filter out the impact of days with low $\Delta TB_i$ caused by cloudy days or synoptic weather systems. The days after or before $\beta$ days will also be checked by Eq. (10) below. In this case, if freezing or thawing state transition, e.g., due to synoptic weather systems, lasts for more than five days, we can still find the annual begins/ends of an FT cycle. Therefore, the influence of synoptic events is excluded.

Then the new algorithm is

$$FT_{new} = \begin{cases} thaw, & if \ \ \text{var}(\Delta TB)_\beta \geq \gamma \ \ or \ \left| \Delta TB_i \right| \geq \gamma \\ frozen, & if \ \ \text{var}(\Delta TB)_\beta < \gamma \ \ and \ \left| \Delta TB_i \right| < \gamma \end{cases} \tag{10}$$


with $\gamma$ a threshold brightness temperature in terms of both $\left| \Delta TB_i \right|$ (for Instantaneous) and $\text{var}(\Delta TB)_\beta$ (for the synoptic weather scale). For example, sunny days will lead to $\Delta TB_i \geq \gamma$ ; cloudy/slow winds days will be filtered out by $\text{var}(\Delta TB)_\beta \geq \gamma$ because these days do not last longer than the period of an atmospheric Rossby wave period. We get $\gamma = 8K$ by statistically computing $\left| \Delta TB_i \right|$ and $\text{var}(\Delta TB)_\beta$ over the northern hemisphere to keep 95% confidence for cases where $T_{2m} < 0°C$ (Figure 2). Any day

that can get $\left| \Delta TB_i \right|$ from SMAP will be checked by Eq. (10). For a day that $\left| \Delta TB_i \right|$ is not available, it will be filled with an FT value depending on the nearest $FT_{new}$.

Figure 2: $\text{var}(\Delta TB)_\beta$ over the northern hemisphere where 95% of samples are within 8K.

## 2.4 Evaluation of the new FT-state detection algorithm

By Eq. (10), one can compute the starting and ending times of the frozen soil period in winter, i.e., the first/last Freeze state in

an annual FT cycle. By applying Eq. (9), it requires at least one FT value per day which affects accuracy in the low-latitudes.

Before a comparison with the half-daily SMAP FT products, we have to scale it to daily resolution by

$$FT_{SMAP-daily} = \begin{cases} thaw, & if \ \ FTstatus_{6am} = 0 \ or \ FTstatus_{6pm} = 0 \\ frozen, & if \ \ FTstatus_{6am} = FTstatus_{6pm} = 1 \end{cases} \tag{11}$$





Eq. (11) produces a bias towards thawed states but matches the concept of the new FT algorithm because $\left|\Delta TB_i\right|$ would be smaller when the state at 6 am and 6 pm are consistent. Hence, the agreement is defined as the fraction of days in which the new method and SMAP FT have the exact daily FT state inference against the total number of days (see Eq. (11)). In detail, both the SMAP's FT product and the new algorithm contain 964x203 grids in the latitude between the equator to 83.6320°N and 2148 days after March 31, 2015. Instead, "agreement" computes the percentage of pairs (one from new FT and the other from SMAP) that are consistent along longitude/latitude/time. SMAP also adopts a similar FT product calibration/validation method as Eq. (11) (Dunbar et al., 2020). The difference is that SMAP needs to compare at 6 am/pm instead of daily scale.

## 3. Results

### 3.1 The performance of the new method at the Xilinhot site

Figure 3a shows the in-situ air and soil temperatures and the SMAP $TB$ observed at or near the Xilinhot site from March 2015 to March 2021. The green lines are the beginnings and ends of the annual freezing cycles inferred from the new FT algorithm. The intervals between green lines in summer indicate the thawed state; otherwise, for the frozen state. In the thawed soil state, $TB$ ranges between 240 K to 280 K. The seasonal $TB$ amplitude follows more or less the variation of air, skin, and upper soil temperatures without phase delay. Under frozen conditions, $TB$ does not exhibit a clear seasonal variation because of a more stable soil emissivity. Figure 3b proves the inference from Eq. (8): $\left|\Delta T_{skin}\right|$ is the maximum difference between the 6 pm. and 6 am. calculated from skin temperature $(T_{skin})$ in the ERA5-land (Fig. 3a). $\left|\Delta T_{skin}\right|$ is relatively stable, about 10-15 K through the years, while $\left|\Delta TB_i\right|$ in winter is two times smaller than in summer.

Figure 3: a) time-series of 2-meter air temperature $T_{2m}$, skin temperature $T_{skin}$, soil temperature at 0-7 cm $stl1$, 7-28 cm $stl2$, 28-100 cm $stl3$, 100- 189 cm $stl4$, as well as SMAP $TB$ at the Xilinhot site; b) time series of $\left|\Delta TB\right|$ computed from the SMAP $TB$s and $\max\left|(\Delta T_{ERA5-land})\right|$. The vertical green dashed lines indicate the beginning and ending day of the soil frozen state as inferred from the new FT algorithm. The gray shades the frozen state from SMAP FT product.

The $\Delta TB_i$ time series in (Fig. 4) shows mostly a substantial intra-annual variation in summer from -40 K to 40 K due to soil moisture variations, which is mainly influenced by the precipitation connected to the East Asia Monsoon. In winter, $\Delta TB_i$ varies only from -8 K to 8 K. However, there are $\left|\Delta TB_i\right|\leq 8K$ is summer as well. To filter out these isolated small $\left|\Delta TB_i\right|$ cases in summer, $\mathrm{var}(\Delta TB)_\beta$ is constructed as in Eq. (9). In Fig.4, $\mathrm{var}(\Delta TB)_\beta$ is close to zero K, but ranges from 0 to 200 K in summer. By comparing with $\Delta TB_i$, $\mathrm{var}(\Delta TB)_{\beta=7}$ (the red line in Fig. 4) successfully filters out the small $\Delta TB_i$ of summer cases. Fig. 4 also shows the sudden changes of both $\Delta TB_i$ and $\mathrm{var}(\Delta TB)_{\beta=7}$ in the beginning and ending time of the frozen soil period according to the new algorithm.

Figure 4: An illustration of diagnosis example at Xilinhot site. The time series of $\Delta TB$ (Eq. (7)) in black line from April 2015 to March 2021. The red line is its seven-day moving variance as in Eq.(9). The green dashed lines are the beginning/ending of the soil frozen state inferred from the new FT algorithm.



At the beginning and after the end of an annual frozen soil period (Fig. 4) $TB_{ih\_6pm}$ is often less than $TB_{ih\_6am}$ (valley in
the time series) while the soil and air temperatures have an opposite behavior due to solar heating during daylight hours. Days
with $TB_{ih\_6pm} < TB_{ih\_6am}$ can be explained by daytime melting of the uppermost few centimeters of the soil, which reduces the
topsoil emissivity. According to Su et al. (2020), intra-daily FT state transitions between the completely thawed and frozen
soil state may last for tens of days at the Maqu site on the Tibetan plateau (Su et al., 2020). The new FT detection algorithm
takes this peculiarity during the transition phase into account and identifies the FT state of the bulk soil rather than of a thin
surface layer as detected by the existing $TB$-based algorithms. Moreover, the new FT detection algorithm requires no local
reference values. Figs. 5 and 6 show, respectively, the beginnings between August and December and endings between January
and May of the frozen soil periods from 2015-2021. For most of the northern hemisphere, there are hardly any frozen soil
states in the northern hemisphere during June and July.

Figure 5: The beginning of the frozen soil periods of the six winters as inferred by the new FT algorithm .

Siberia and Tibet experience the earliest soil freezing. In East Asia, the beginning of soil freezing follows the latitude and
the East Asia Summer Monsoon propagation in a meridional direction. Regions with an earlier retreat of the East Asia Summer
Monsoon also freeze earlier, i.e. the region reaching from Mongolia to southern China show in November a northwest-
southeast freezing beginning gradient. Over Europe, the gradient direction of the start times exhibits a southwest-northeast
pattern towards Siberia. The freezing begins about two months later than in other regions with the same latitude (40°N~60°N).
In North America, the freezing starts in October in the north and invades the south in December. Winter 2015-2016 and Winter
2017-2018 show more severe soil freezing than other winters. Deep blue colors along 120°W mark the Rocky Mountains.

The pattern of the frozen soil ending dates (Fig. 6) is similar to the one of the starting dates. The frozen soil period lasted
longest in Siberia, Tibet, and the Rocky Mountains. Winter 2017-2018 has the latest ending time of the frozen soil period in
North America and Europe. The contours of the frozen soil period's beginning vary by tens of days from year to year. Parts of
Siberia and Tibet never have unfrozen soil (deep blue colors); the soils are also frozen during June and July, which are not
shown in the figures. Color transitions in Fig.6 are more smooth and continuous than Fig. 5 and show a more gentle date
gradient. For instance, the gradient over Eurasia is spotted spatially in Fig. 5. An air temperature drop may explain the jagged
color contours in Fig. 5. These delicate patterns reflecting topography, sea-land distribution, and climatological types are not
observed in Fig. 6, implying a more gradual and slow heating process in spring.

By combining Fig.5 and 6, we get the duration of the annual frozen cycle as in Fig. 7. The most extended frozen soil
period (>200 days) is located near the North Pole and over some regions in Tibet (Fig. 7). The sharpest gradient in the length
of the frozen period stretches from the eastern part of Siberia to the northeast part of China following the latitudes; here, the
freezing duration decreases from more than 300 days to 90 days in about ten days per latitude degree. The freezing duration
does not change much between the years and exhibits similar patterns.

Figure 6: The ending of the frozen soil periods of the six winters as inferred by the new FT algorithm.

Figure 7: The duration of the frozen period inferred from the SMAP L3 global H-polarization brightness temperature product.

**3.2 The comparison with the SMAP FT products**





In this section, we will compare the results from the new FT algorithm with SMAP's regarding time variation and spatial distribution. 2-m air temperature will also be adopted as a reference and be compared with both FT results.

Table 1: $FF_{fr}$ and $FF_{th}$ from SMAP's algorithm at Xilinhot site.

| summer | 2015 | 2016 | 2017 | 2018 | 2019 |
|---|---|---|---|---|---|
| $FF_{th}$ | 0.1085 | 0.1118 | 0.1078 | 0.1095 | 0.1074 |
| winter | 2015-2016 | 2015-2016 | 2015-2016 | 2015-2016 | 2015-2016 |
| $FF_{fr}$ | 0.0251 | 0.0277 | 0.0302 | 0.0274 | 0.0295 |


Fig. 8 shows the zonal average agreement evolution along in the annual FT transition zones between 30°N-80°N. Although the SMAP FT product contains the data in 0°N -30°N, the zones that are supposed to be freeze-free or permanently frozen covered (>83°N) are not included in Fig 8. The new FT detection algorithm results mostly agree well with the SMAP FT product (Fig. 8). In a transition zone, the overall agreement is only 0.6-0.7. The agreement of the detected beginning and ending times in the 30°N-40°N latitude belt is lower than the zones above 40°N because of the Tibet Plateau (25°N-40°N, and see Fig 9a). The agreement between the two algorithms can be below 0.5 in winter (the blue area in Fig. 8, especially ≥80°N).

In summer, both agree up to 0.98 and 0.70-0.75 in the wintertime. However, the lowest agreement does not happen in the deep winter but some time ahead and after January by showing the two valleys as seen for each year in Fig. 8. These valleys are the freezing/thawing transition period in the northern hemisphere that gets the lowest agreement. Thus, the primary disagreement between SMAP's threshold=0.5 and the new one's $\gamma = 8K$.

Figure 8: The fraction of agreement time series along the latitudes.

The difference in Fig. 8 is due to the hypothesis in both algorithms. For the SMAP FT algorithm, it requires $FF_{fr}$ and $FF_{th}$; their estimate needs sufficiently long periods of wet and frozen soil, which becomes increasingly difficult with decreasing latitude. Thus, $FF_{fr}$ and $FF_{th}$ can be unreliable for the transition zones where the frozen emission character is not typical and may have interannual changes. For instance, the FFfr and FFth change variation can reach 4% and 20%, respectively, and strongly depend on the samples. Table 1 illustrates $FF_{fr}$ and $FF_{th}$ variation at the Xilinhot site by adopting different samples every year. Another reason for the difference in Fig 8 is the assumption adopted by the new algorithm. For one side, the new algorithm cannot identify a frozen soil period shorter than seven days (as shown in Eq. (9)). It leads to the thawed/frozen error in the new FT algorithm where the 3-5 days heat/cold events may result in different FT states from the two algorithms.

Figure 9: The spatial pattern of the fraction of agreement in the northern hemisphere between a) the new method and SMAP L3 FT state products; b) the new method and 2 m air temperature; c) the SMAP L3 FT state products and 2 m air temperature ; d) the difference by b minus c. The red cross marks the location of Xilinhot.


Fig. 9a compares the SMAP FT state product and the new algorithm regarding the spatial distribution of the agreement. The Tibetan Plateau shows the lowest agreement value in Asia, especially its southern margin close to the Himalayas Mountains. While for most of the plateau, the agreement stays above 0.7, it is below 0.5 at some points organized in a belt in an east-west direction. The area downstream of the plateau, including the center parts of China, also shows a strong disagreement between both estimates. A zone with a low agreement (0.7-0.8 ) is found between 50°N-60°N, especially the





Lake Baikal region, northern Europe, and the east coast of Canada and Alaska. A zone with a high agreement is found in eastern China, central Asia, southwestern Europe, and the southern U.S. Fig 9b&c make the same agreement map between the new/SMAP's FT algorism and the FT stated inferred from $T_{2m}$ by a binary judgment of the freezing point (273.15K), i.e., $T_{2m}<273.15K$ is frozen and $T_{2m}>273.15K$ is thawed. Fig 9b&c shows that despite some mismatch between the results of the
new algorithm and the SMAP FT, both overall agree by more than 70%. The new FT state detection better agrees with $T_{2m}$ in the mid-latitudes than the SMAP FT algorithm but worse in latitudes above 60°N and low latitudes below 30°N.

## 3.3 Sensitivity test

Parameter $\beta$, and $\gamma$ in the new FT state detection have been selected based on the typical length of synoptic weather systems and the 95% confidence level outcome. $\beta$ (days) is the window length (days) over which the variance in Eq. (9) is estimated
and used as a decision criterium in Eq.(11), which filters out sporadic changes by weather events. $\gamma$ is a threshold temperature to judge if the observed SMAP TB signal is related to frozen soil in Eq. (10). The values for these parameters have been set in an ad-hoc fashion; here, we analyze the sensitivity of the results against the soil freeze/thaw state inferred from $T_{2m}$ to the variation of these parameters by varying, $\beta$ from 5 to 11 K (Figs. 10a, 11a&b), and $\gamma$ from 3 to 11 days (Figs. 10b, 11c&d).

Figure 10: The time-series of the fraction of agreement between the new method and the FT state inferred from $T_{2m}$ detection in the northern hemisphere regarding a) $\beta$, the window length for the variance; b) $\gamma$, the threshold temperature difference.

305       When only changing one parameter, the degree of agreement between the new FT estimates and $T_{2m}$ changes significantly only in winter (Fig. 10). The impact of changing only $\beta$ is also larger in winter than in summer (Fig. 10a). A smaller $\beta$ reduces the agreement from above 0.73 (at $\beta=7$) to 0.65 when $\beta=3$, while a larger $\beta$ only barely increases the agreement between both estimates. A smaller $\gamma$ improves the agreement in winter (Fig. 10b) by more than 0.1. Overall, the variation of the three parameters leads only to significant changes in the agreement between both estimates in the thawing period. The spatial
distribution of the change of agreement between both FT state estimates relative to the default values for the three parameters is shown in Fig. 11. The Tibetan Plateau and some Northern Europe areas behave opposite to Siberia, the western coast of North America, and the region around latitude 30°N.

Figure 11: The spatial pattern of the fraction of agreement between the new method and the FT state inferred from $T_{skin}$ detection in the northern hemisphere for, β=3 (a), β=11 (b), γ=5 (c), and γ=11 (d).

## 4. Discussion and conclusions

We developed a new FT state detection algorithm based on the difference of the microwave brightness temperature between 6 am descending and 6 pm ascending half-orbit passes that are relatively small over frozen soil due to the large penetration depth leading to an effective temperature dominated by stable deeper soil temperatures. The new FT state detection agrees well with SMAP's FT state detection with a minimum of above 0.72 in winter. The new algorithm can reach a comparable agreement regarding the spatial distribution as the SMAP FT product does against $T_{2m}$ FT inference in the northern hemisphere. The
algorithm is rather stable to variations of its ad-hoc set parameters. The new algorithm presented in this study is expected to extend the application of L-band passive microwave remote sensing data in freezing-thawing conditions.

      The primary issue of the FT product calibration/validation of passive microwave remote sensing FT products is how to measure and define the FT on the ground, especially for a deeper penetrated band like the L band. We lack a precise "ground truth" of soil's freeze/thaw state. $T_{2m}$ FT inference is used in this study, and SMAP uses a more complicated scheme by





considering $T_{2m}$, $T_{skin}$, $T_{5cm}$, and $T_{10cm}$. Since in-situ $T_{2m}$ is usually to either infer the ground FT states or used in the FT products calibration/validations, it is interesting to know the relationship between air temperature and FT state on a global scale. However, when it comes to soil temperatures like $T_{skin}$, $T_{5cm}$, $T_{10cm}$ or more profound, it is hard to say which layer can correspond to the $TB$ signal best. Another option is in-situ soil temperature at 5 cm as in SMAP's FT product calibration/validation. However, either the detectable depth or the footprint of SMAP's radiometer does not match ERA5-land.

Indeed, the soil temperature from a particular layer can not compare with SMAP's FT product because the penetration depth of the L-band signal is dynamic, especially for the frozen soil that may range from a few centimeters to meters. Thus, the comparison in this study, such as Fig. 9-11, is hard to be considered as an evaluation of the new FT algorism or SMAP's but a comparison to show that the new FT algorithm can capture FT signals as good as the SMAP's official one.

Figure 12: The cover of the frozen land detected by the new FT algorithm and $T_{2m} < 273.15K$.

Since $T_{2m}$ is adopted in this study, Fig. 12 shows the frozen land cover and annual accumulations inferred from the new FT state algorithm and the thermal conditions near the surface from $T_{2m}<273.15K$ in the northern hemisphere. Both are very similar, while the maxima reached for both data sets are different due to their different grid sizes; SMAP data are given in area-equal grids 36 km in diameter (about 1300 km$^2$), while $T_{2m}$ used a 1°x1° lat-lon grid (0 to 1000 km$^2$). However, the frozen land area detected by the new FT algorithm is about 15 days in advance of $T_{2m}$, especially in spring (the shift of black lines).

It coincides with the challenges in using $T_{2m}$ as validation for L-band derived estimates of FT since the radiometer measurements are sensitive to the near-surface soil layer. The time lag in springs is reasonable because soil absorbs the solar radiation and then heat air temperature. The accumulated frozen land areas are also different. Usually, the air temperature will lead to more frozen soil flags except for the year 2019. Fig. 12 shows that $T_{2m}$ is more appropriate to be "ground truth" in the beginnings than the endings of an annual FT cycle. A similar issue exists for $T_{skin}$, $T_{5cm}$, $T_{10cm}$ or more profound layers. The

penetration depth of the L-band detection varies severely during the FT transitions; thus, it would not be suitable to evaluate the FT algorism with a static depth.

Thus, clarifying the "ground truth" is critical to developing FT remote sensing algorism at the L-band. The future work needs more in-situ calibration/validation activities in the field experiment, not only for the satellites but from tower-based or airborne platforms. The L-band TB detected by the satellites covers the FT states in tens of kilometers, and most importantly,

the FT has a vertical structure. The field experiment is expected to identify the sensing depth, which is also a vague concept for soil moisture remote sensing at the L-band during the freezing/FT transition periods.

**Acknowledgments**

This research was funded by the National Natural Science Foundation of China (Grant 42075150), and by the Natural Science

Foundation of Shanghai (No. 21ZR1405500). Also by the Deutsche Forschungsgemeinschaft (DFG) via the research group FOR2131 on "Data Assimilation for Improved Characterization of Fluxes across Compartmental Interfaces", subproject P2.

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



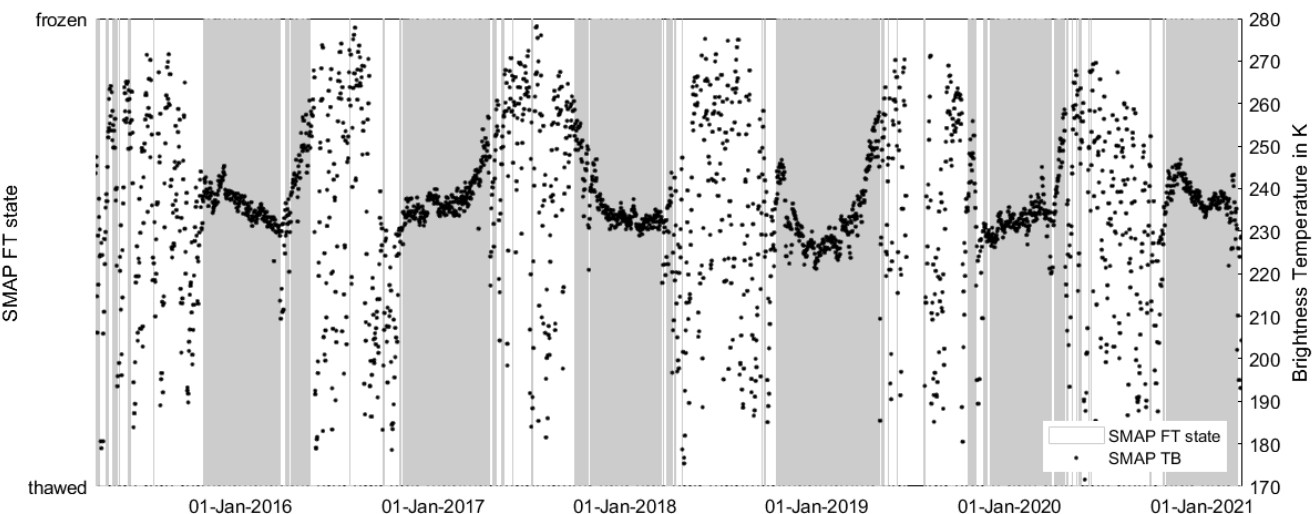

Figure 1: SMAP *TB* time series and the derived reference FT state (grey for frozen and white for unfrozen) extracted for the location of the Xilinhot site.

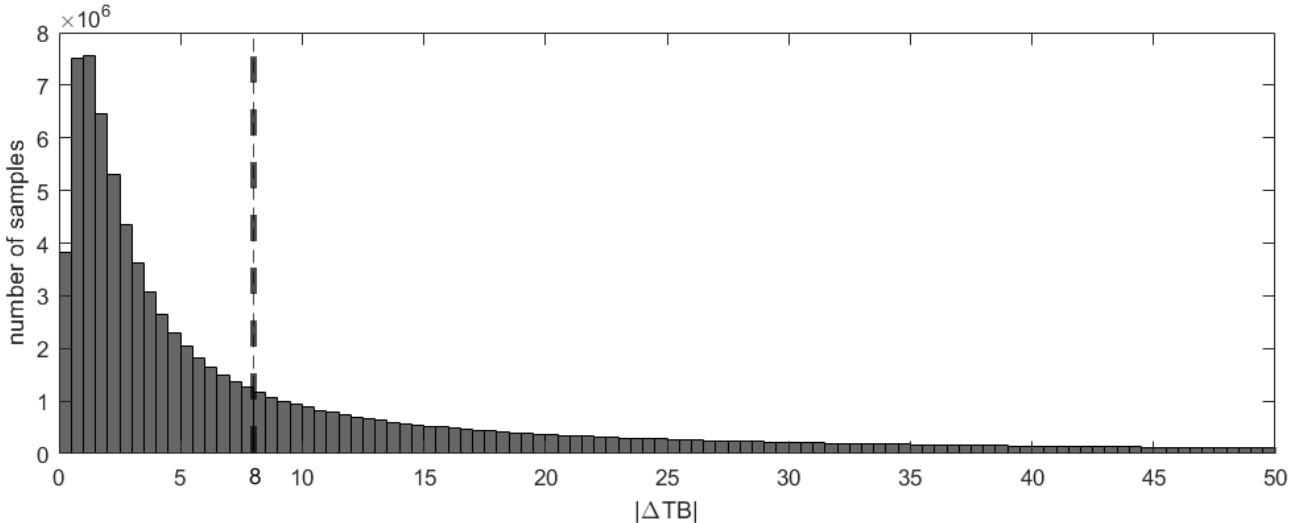


Figure 2:  var$(\Delta TB)_\beta$ over the northern hemisphere where 95% of samples are within 8K.

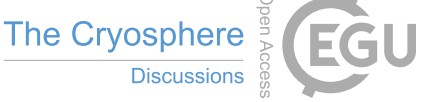

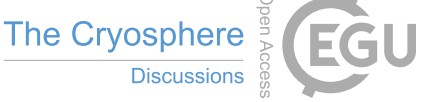

Figure 3: a) time-series of 2-meter air temperature $T_{2m}$, skin temperature $T_{skin}$, soil temperature at 0-7 cm $stl1$, 7-28 cm

$stl2$, 28-100 cm $stl3$, 100- 189 cm $stl4$, as well as SMAP $TB$ at the Xilinhot site; b) time series of $|\Delta TB|$ computed from

the SMAP $TB$s and $\max|(\Delta T_{ERA5-land})|$ . The vertical green dashed lines indicate the beginning and ending day of the soil

frozen state as inferred from the new FT algorithm. The gray shades the frozen state from SMAP FT product.



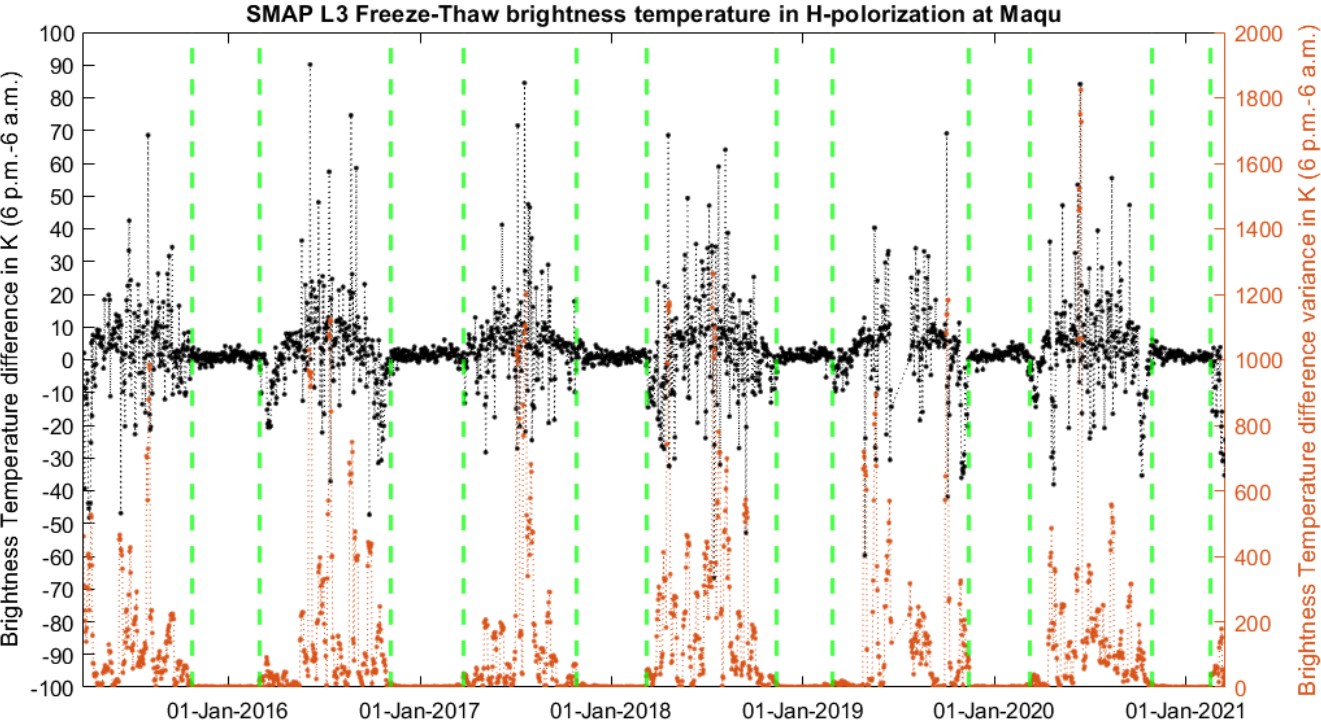

Figure 4: An illustration of diagnosis example at Xilinhot site. The time series of $\Delta TB$ (Eq. (7)) in black line from April 2015 to March 2021. The red line is its seven-day moving variance as in Eq. (9). The green dashed lines are the beginning/ending of the soil frozen state inferred from the new FT algorithm.







Figure 5: The beginning of the frozen soil periods of the six winters as inferred by the new FT algorithm.







Figure 6: The ending of the frozen soil periods of the six winters as inferred by the new FT algorithm.






Figure 7: The duration of the frozen period inferred from the SMAP L3 global H-polarization brightness temperature product.



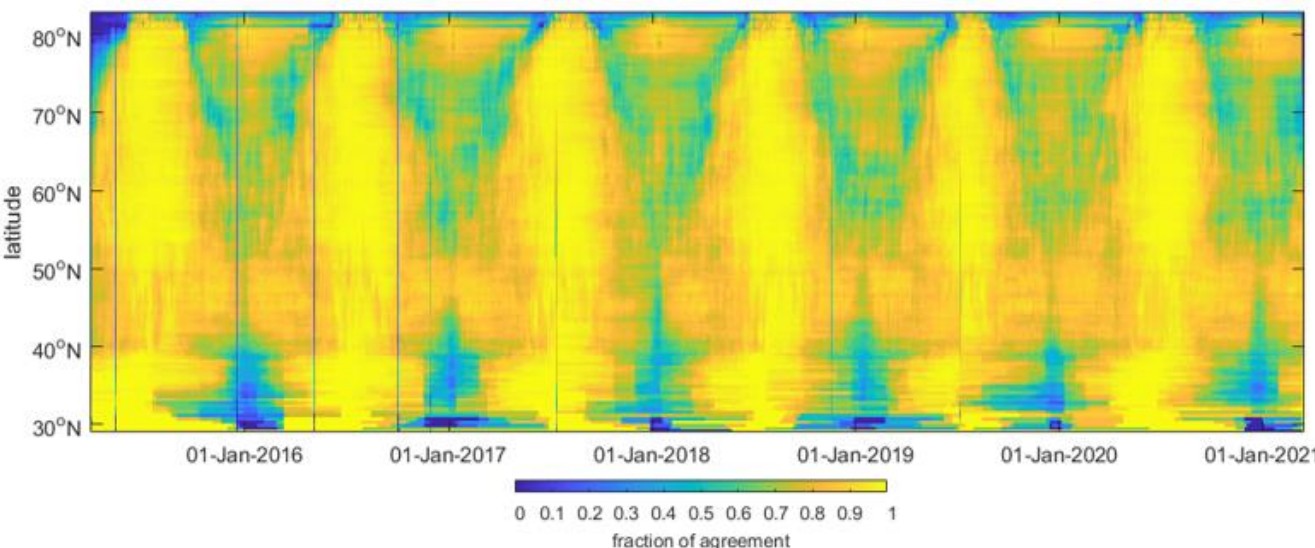

Figure 8: The fraction of agreement time series along the latitudes.





Figure 9: The spatial pattern of the fraction of agreement in the northern hemisphere between a) the new method and SMAP L3 FT state products; b) the new method and $T_{2m}$; c) the SMAP L3 FT state products and $T_{2m}$; d) the difference by b minus c. 510 The red cross marks the location of Xilinhot.





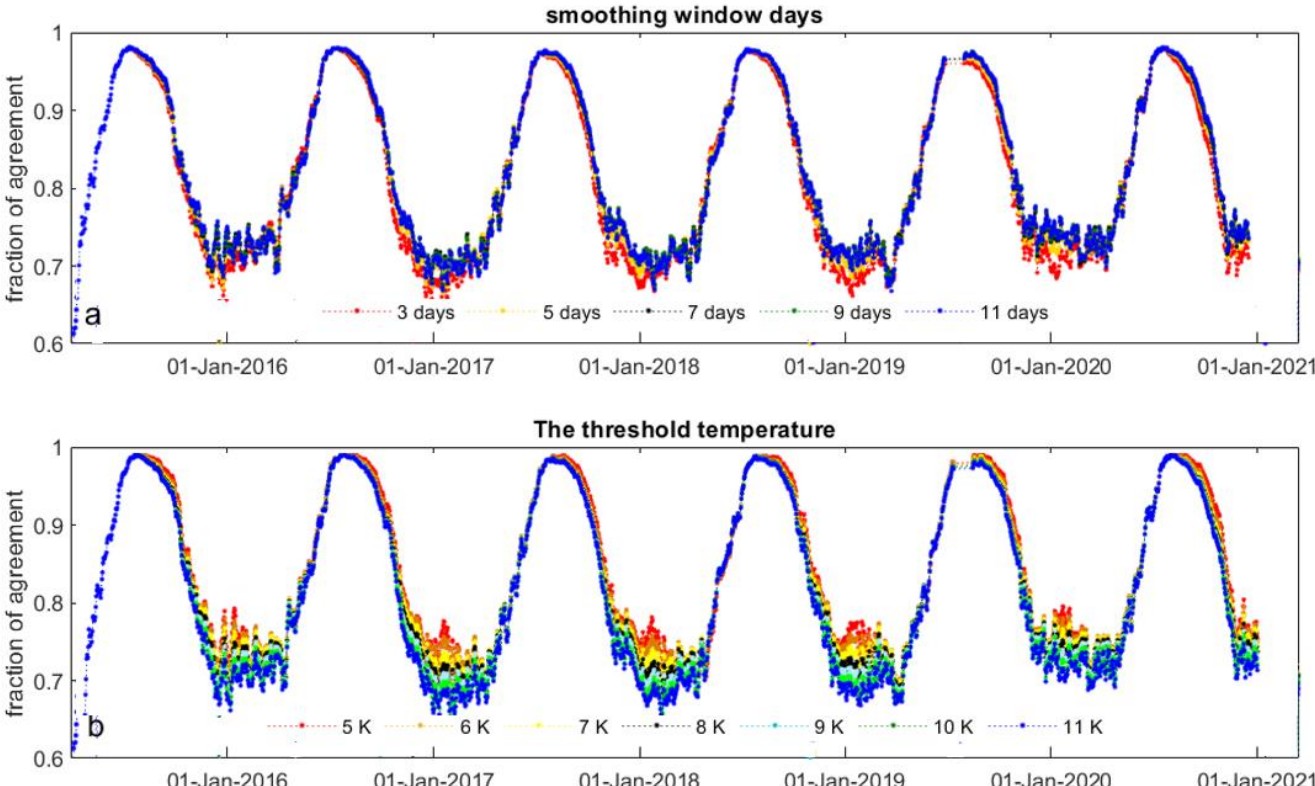

Figure 10: The time-series of the fraction of agreement between the new method and the FT state inferred from $T_{2m}$ detection in the northern hemisphere regarding a) $\beta$, the window length for the variance; b) $\gamma$, the threshold temperature difference.






Figure 11: The spatial pattern of the fraction of agreement between the new method and the FT state inferred from $T_{skin}$ detection in the northern hemisphere for, β=3 (a), β=11 (b), γ=5 (c), and γ=11 (d).



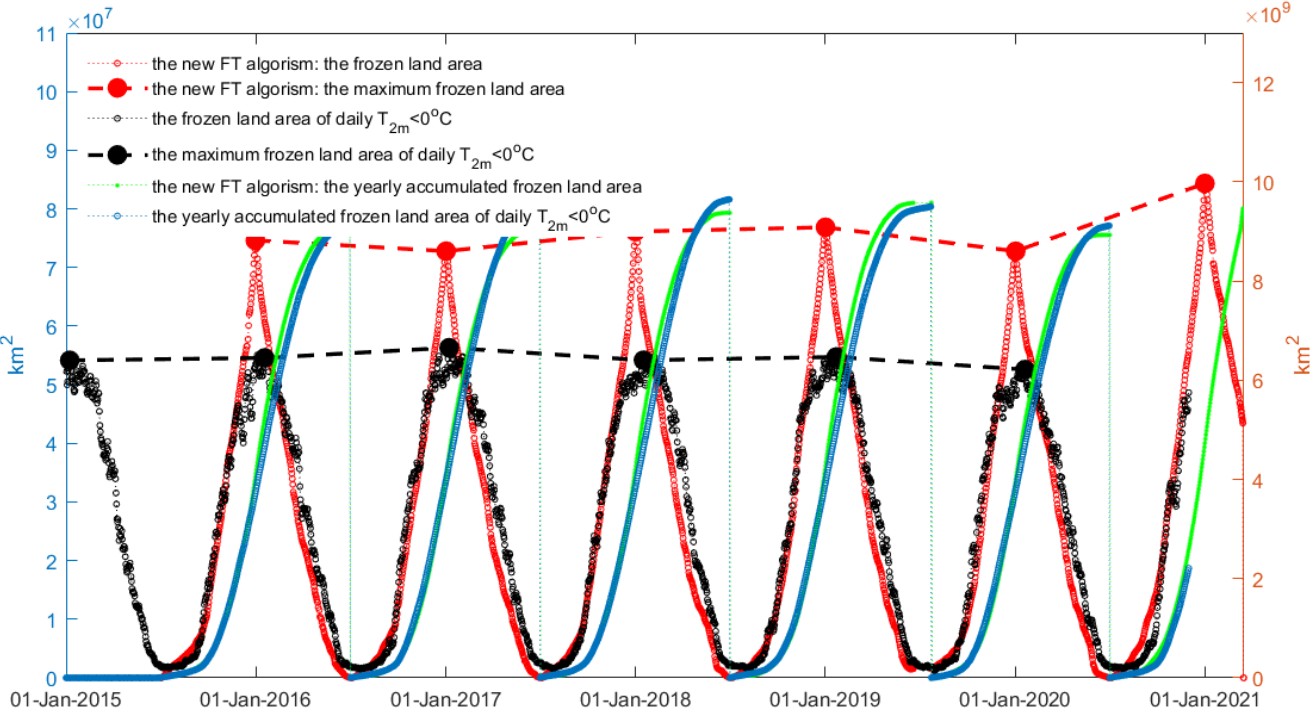

Figure 12: The cover of the frozen land detected by the new FT algorithm and $T_{2m} < 273.15K$.