# Peer review of "A Novel Global Freeze-Thaw State Detection Algorithm Based on Passive L-Band Microwave Remote Sensing"

_The Cryosphere, 2021_

## Author Comment (AC1)

Reviewer #1

Lv et al. present a new remote sensing algorithm for identifying the land surface freeze/thaw state using SMAP passive microwave observations. The central premise of the algorithm is that landscapes that remain frozen on daily to synoptic time scales are characterized by small diurnal differences in brightness temperature. The authors compare the results obtained with their algorithm with the SMAP product and various reanalysis-based freeze/thaw-related temperature indices over the Northern Hemisphere. They further derive common F/T metrics such as the length of the frozen period, but they do not analyze them in depth.

The manuscript's content is relevant to The Cryosphere, as it covers a topic that is of interest to the journal's audience and also to applied remote sensing scientists. However, it suffers from several major weaknesses and inconsistencies that will require extensive revision or rewriting before publication. The main shortcomings are:

1) opaque writing that curtails comprehension (paper structure, poorly structured paragraphs, figures)

2) inconsistencies between the author's claims that their algorithm is globally applicable and the assumptions underlying the algorithm

3) multiple claims that are not backed up by evidence or references

4) limited scrutiny of the algorithm and its output

Reply: Thank you very much for your meticulous comments and feedbacks. We have revised the paper and explained your comments in the following.

1) Lack of clarity

The manuscript is difficult to read. In addition to many poorly worded phrases that detract from the content, the paper structure, the paragraph structure and the figures are challenging to follow.

Reply: Sorry for that. I improved the manuscript point by point.

The paper structure is unusual in that the authors do not include a proper discussion (see below). Furthermore, the introduction does a poor job of conveying the main ideas and findings. For instance, the variance-based filtering that is central to the algorithm is not mentioned. The reader is caught by surprise in the methods section.

Reply: Thanks a lot for this comments. We had made corresponding changes to make this point clear:

"The new FT algorithm has a similar basis to the DAV approach(Sharifnezhad et al., 2021) applied to higher frequency passive microwave measurements for snow and ice sheet applications(Kopczynski et al., 2008; Tedesco, 2007). It has been proved the Durinal Amplitude Variation (DAV) of passive microwave signals are sensitive to the FT state changes(Sharifnezhad et al., 2021), which are dynamic and complex and vary continuously in space and time." is shifted (revised) from Section 2.3 to Line 66-67.

More broadly, I suggest the authors clarify what they mean by the estimand, i.e., the freeze/thaw state. Is it defined instantaneously (with the variance-based filtering just a convenient means to stabilize the estimation), or is it aggregated on daily or synoptic time scales?

Reply: sorry for the unclear statement here. From Eq. 1 to Eq. 8, we took the daily scale as the other DAV approaches (added in Line 183). By adding Eq. 9 and Eq. 10, we add synoptic time scales background to daily values (added in Line 203). Some clarifications were added to make

this point clear.

The paragraphs often appear to be haphazardly put together, thus greatly limiting the readability of the manuscript. The introduction serves as a good example. The paragraph starting at line 47 opens by highlighting the limitations of temperature-based indicators for freeze/thaw state estimation. The second sentence states that "[i]n contrast, more direct state information results from the very different microwave dielectric constant for frozen and unfrozen soil. However, the reader has to guess that this sentence and the paragraph it is contained in are about microwave remote sensing of the freeze/thaw state, as the expressions "remote sensing" and "freeze/thaw" are not mentioned once.

Reply: Thanks a lot for your comments. We'd explicitly state in our revisions in Line 47. "However, FT state estimations from in-situ temperature observations are limited in scale, and it is not straightforward to deduce the state from soil, skin, or near-surface air temperature." is shifted to the previous paragraph. From MEaSUREs to SMAP' FT products, the paragraph is about remote sensing of FT state with the passive microwave remote sensing.

The remainder of the paragraph talks about emissivities without referring to the frequency and polarization.

Reply:   Yes. And "For the L-band." is added in Line 51.

At some point, the reader stumbles upon L-band observations from SMAP and SMOS, which, however, are of central importance to the manuscript and to the introduction. I suggest the authors identify one theme for each paragraph and structure the paragraph such that the reader can easily follow.

Reply: We revise the Introduction part, especially to ensure the first sentence indicates each paragraph's core idea. We hope it is fine for the potential readers now.

Another example of opaque writing is furnished by lines 177-190. The authors first propose their own definition of the beginning/end of "the annual freezing", but the subsequent algorithm is seemingly at odds with the definition. For instance, a brief cold spell in summer would meet the definition but would in most cases be screened by the variance criterion.

Reply: Thank you for this comments. "a brief cold spell in summer would meet the definition but would in most cases be screened by the variance criterion." is exactly how the new algorithm works.

The authors further claim that their variance screening using a window length beta of 7 (implicit unit: days) "optimally" filters out brief events. However, it is not at all clear how optimality is defined and what the evidence is that optimality is achieved.

Reply: Thank you for your suggestion. We agree that "optimally" is not appropriate here because Figure 9&10 already shows that it depends on the location and scale. "optimally" is deleted from the text.

The figures are exceedingly difficult to interpret. For example, figure 3 is a cornucopia of lines and markers, with poor contrast (the yellow line is almost invisible) and most items being obscured by others that are plotted on top.

Reply: As suggested by another reviewer, Figure 3a removes soil temperature time-series.

The captions contain insufficient information to interpret the figures. For instance, Fig. 11 does not explain how the fraction of agreement (negative in the figure) was computed. Is it a difference?

Reply: thank you. The caption is revised as "Figure 11: The spatial pattern of the fraction of

agreement difference compare to Figure 9b for β=3 (a), β=11 (b), γ=5 (c), and γ=11 (d)."

The caption of figure 2 shows a histogram of the beta-windowed variance, but it is not explained what input data were used and what value of beta was used.

Reply: Thanks a lot. The caption is revised as " $\mathrm{var}(\Delta TB)_{\beta}$ with β=7 by SMAP TB data

contained in SMAP L3 Radiometer Global Daily 36 km EASE-Grid Freeze/Thaw State, data over the northern hemisphere where 95% of samples are within 8K."

2) Globally applicable algorithm?
The authors claim in the title that their algorithm is globally applicable, but the limited applicability of the underlying assumptions casts doubt on this claim. The authors do little to dispel these concerns, as they do not include a separate discussion section where associated limitations ought to be scrutinized. I also note that despite the word global in the title, no results for the southern hemisphere are provided.

Reply: we remove the word "global" from the title. This is the only place we refer the new algorithm's adapbitlity. However, we have to note that all sites in SMAP handbook (Algorithm Theoretical Basis Document (ATBD) SMAP Level 3 Radiometer Freeze/Thaw Data Products (L3_FT_P and L3_FT_P_E)) for FT Cal/Val are taken from the northern hemisphere, and SMAP provide global FT products.

However, my two biggest concerns in this respect are the assumption of 6 am / 6 pm overpasses and the assumption of elevated variability of the dielectric characteristics of thawed landscapes on synoptic scales. Neither of these two assumptions are very accurate on a global scale.The assumption of 6 am / 6 pm overpasses is difficult to defend at high latitudes, where the temporal sampling deviates substantially from that at the equator. The authors neglect this issue completely, although negative repercussions on their algorithm's performance are not too difficult to imagine.

Reply: it is to note that the 6 am / 6 pm overpassing time are not assumptions. In SMAP hand book (Algorithm Theoretical Basis Document (ATBD) SMAP Level 3 Radiometer Freeze/Thaw Data Products (L3_FT_P and L3_FT_P_E)) on page 26 and other places, it is said "The SMAP L3_FT_P product distinguishes 4 levels of freeze/thaw conditions determined from the ascending (6AM) and descending (6PM) orbit retrievals, including frozen (from both AM and PM overpass times), non-frozen (AM and PM), transitional (AM frozen; PM non-frozen) and inverse-transitional (AM non-frozen; PM frozen) states". In all cases, the daily L3_FT products incorporates (AM and PM satellite overpasses) data for the current day, as well as past days information (to a maximum of 3 days, necessary only near the southern margin of the FT domain) to ensure complete coverage of the FT domain in each day's product.

The assumption of elevated variability of the dielectric characteristics of thawed landscapes on synoptic scales is not subjected to any scrutiny.

Reply: NASA MEaSUREs already can distinguish FT heterogeneity in accordance with mesoscale climate and landscape, please refer to http://www.ntsg.umt.edu/freeze-thaw/
Kim, Y., Kimball, J. S., Glassy, J., and Du, J. Y.: An extended global Earth system data record on daily landscape freeze-thaw status determined from satellite passive microwave remote sensing, Earth System Science Data, 9, 133-147, 10.5194/essd-9-133-2017, 2017.

The authors acknowledge that the Rossby wave time scale that serves as foundation for the beta parameter is relevant to mid-latitudes, but they do not discuss their variance-based

filtering within a time window of length beta affects the results elsewhere.

Reply: Thanks for Reviewer's comments. We believe this is not the case as perceived by the reviewer. Figure 10a discusses how beta affects the domain(lat>20°N). If elsewhere means a particular place, please refer to Figure 11. Among the regions of particular concern, we list cold arid regions (mentioned by the authors as presenting challenges to microwave F/T algorithms in the introduction), bedrock-dominated areas, and regions with extended periods of stable anticyclone.

3) Unsubstantiated claims

The authors make numerous claims that are not backed up by evidence or references. An excellent example is furnished by the paragraph starting on line 153, whose intent is to provide a rationale for the new algorithm. There, the authors make numerous such claims. For instance, they state that brightness temperature changes during freeze/thaw transitions are at least as large as those associated with precipitation "because of the huge epsilon difference between frozen and unfrozen soil". They do not provide a reference or evidence for this claim, nor do they state when this may not be the case (e.g., certain arid landscapes).

Reply: Thank you. Please refer to Sharifnezhad et al., 2021 with a DAV signals analysis. And we'd added this citation in the revision.

Sharifnezhad, Z., Norouzi, H., Prakash, S., Blake, R., and Khanbilvardi, R.: Diurnal Cycle of Passive Microwave Brightness Temperatures over Land at a Global Scale, Remote Sensing, 13, 817, 2021.

A further issue is that the language is inappropriate and vague ("huge"). There are numerous similar claims in this paragraph alone, and not a single piece of evidence or reference is provided.

Reply: Thanks a lot. We try to amend throughout the manuscript and change the words like "huge" to an appropriate phrase.

4) Very limited scrutiny

The authors do not subject their algorithm and its underlying assumptions to the level of scrutiny that a reader of The Cryosphere may expect. There is no discussion section that assesses failure cases or that establishes a link between potentially inappropriate assumptions and questionable results.

Reply: Thanks a lot for this comment. We try to present such scrutiny via formulas and numbers in Section 2.3. Besides, we're trying to repeat the work with SMAP's core site data for SMAP's FT products. The accuracy in SMAP's FT products is expected to be 80%, and some studies show that it is >70%. The new algorism is not validated because of the reasons explained in Discussion, i.e., lack of ground truth. What we did is to prove that the new algorithm does relate to the FT state transition in theory, and it has comparable accuracy with SMAP's. We now also include some of limitations of our theory in the discussion in Line 369-376.

Furthermore, general issues with the "validation" strategy employed here (e.g., scale and commensurability with reanalysis-derived temperature metrics) should be incorporated.

Reply: Thank you for your suggestion. We add, "The SMAP FT team uses WMO's air temperature, and WMO's air temperature is still sparse for the agreement assessment. For instance, how to deal with the scale mismatch between the weather station and SMAP's

footprint? How to account for sub-grid open water fraction, terrain heterogeneity, tree cover, precipitation, and snowmelt and on with the weather station data? These problems can be avoided for ERA5-land air temperature in the evaluation, and we are aware that the ERA5-land air temperature is not appropriate for validation which needs FT ground truth for sure." In Line 325-328.

Minor comments

l 37: suggest replacing solar with shortwave and terrestrial with longwave

Reply: These words are updated.

l 40: Potentially inappropriate reference (Schuur et al.): How does the surface freeze/thaw state relate to permafrost carbon

Reply: as mentioned in P174 as "A number of ecosystem and Earth system models have incorporated the first approximation of global permafrost carbon dynamics. Recent key improvements include the physical representation of permafrost soil thermodynamics and the role of environmental controls, in particular the soil freeze/thaw state, on a decomposition of organic carbon". We replace it with

44. Koven, C. D., Riley, W. J. & Stern, A. Analysis of permafrost thermal dynamics and response to climate change in the CMIP5 Earth system models. J. Clim. 26, 1877–1900 (2013).

45. Koven, C. D. et al. Permafrost carbon-climate feedbacks accelerate global warming. Proc. Natl Acad. Sci. USA 108, 14769–14774 (2011).

l 94: "replaying": odd choice of word

Reply: We refer to https://www.eea.europa.eu/data-and-maps/data/external/era-interim-1 for the origins of "replaying".

l 171: That \Delta TB_i will be smaller than \Delta T_i does not follow from the provided inequalities because (7) is a sum. A mathematically sound argument is needed to substantiate the claim.

Reply: (7) is not just a sum but a sum with weighting function. The sum of the weightings, i.e.,

$\left(1-e^{-\tau_1}\right)<1$, $\left(1-e^{-\tau_i}\right)\prod_{j=1}^{i-1}e^{-\tau_j}<1$ and $\prod_{j=1}^{n-1}e^{-\tau_j}<1$, is equal to 1.

---

## Author Comment (AC2)

Reviewer #2

Review on A novel global freeze-thaw state detection algorithm based on passive L-band microwave remote sensing, by Lv et al., (tc-2021-369).

This paper used Diurnal Amplitude Variation (DAV) to detect the landscape FT status over Northern hemisphere using SMAP L-band H-pol brightness temperatures. The performance of the FT classification was assessed using ERA5 2m air temperature and other global SMAP FT data records. The paper covers a topic that is suitable to readers of The Cryosphere and should be of particular interest to those interested in FT classification algorithm development and FT dynamics under climate change. However, the manuscript has concluded with lack of detail in describing method and FT classification algorithm, and insufficient FT agreement assessment. Additional analysis on relationship between L-band signal and soil temperature should be added to improve the conclusion (See "line 329-331" below). The suggested major revisions are as follows:

- Major concern is FT agreement assessment. Authors used air and skin temperatures, and soil temperatures at several depths from a single site (Xilinhot). Agreement assessment from only one site is not enough for global scale FT validation. FT sensitivity to L-band Tb signal varies on land cover type and climate regions.

Reply: Sorry for the misunderstanding. Agreement assessment (Section2.4) is achieved in the domain latitude > 20°N by ERA5-land and SMAP FT products (both TB and F/T information contained). We did not calculate the agreement at the Xilinhot site. The role of the Xilinhot site is to illustrate the assumption/algorithm of the new method described in Section 2.3 at the Xilinhot site. We revise the subtitle of Section 2.3.

- In the accuracy agreement at global domain, ERA5 is a model reanalysis data with uncertainty as well. Authors should include additional global FT agreement assessment instead of using only ERA5 data.

Reply: We agree that more assessments should be taken, and ERA5-land is just one reanalysis product. As we mentioned in the Discussion, the lack of ground truth is vital to developing the L-band remote sensing FT products. To acquire precise soil temperature/air temperature data is beyond the scope of this study. The SMAP FT team uses WMO's air temperature, and WMO's air temperature is still sparse for the agreement assessment. For instance, how to deal with the scale mismatch between the weather station and SMAP's footprint? How to account for sub-grid open water fraction, terrain heterogeneity, tree cover, precipitation, and snowmelt and on with the weather station data? These problems can be avoided for ERA5-land air temperature in the evaluation, and we are aware that the    ERA5-land air temperature is not appropriate for validation which needs FT ground truth for sure. Similar words are added in Line 331-334.

- Additional analysis on relationship between L-band signal (FT dynamics as well) and soil temperature should be added to improve the conclusion. That would be the possible reason why L-band microwave remote sensing can be used for better penetration depth monitoring.

Reply: The penetration depth issue and sensing depth issue for soil moisture retrieving from L-band is still not clear (Lv et al. 2019 on IEEE TGRS). The problem is more complex for frozen soil because the dielectric profile is not continuous if freeze-thaw transitions happen in the middle layers of a profile, and a complete frozen soil profile shall have much deeper penetration theoretically. Thus, we can not even get a precise penetration depth for the case

of a    freeze-thaw transition. By using the air temperature, we avoid this complicated situation. Otherwise, to select which layer and according to what standard in comparison with the SMAP FT products would be vulnerable. In this study, we use the same method as the SMAP handbook, which is considered as the state-of-art in this topic, to get the agreement.

- Although this study provided better overall FT classification accuracy, it is not clear that what factors (or which land cover type?) contribute to improve FT classification accuracy or degrade. Other landscape factors affect FT classification accuracy. The factors include sub-grid open water fraction, terrain heterogeneity, tree cover, precipitation and snowmelt and on. To improve the quality of the paper, additional analysis and discussion on this should be required.

Reply: Thank you for your suggestion. We admit the method is not perfect, but seeing from the result, the agreement (Figure 9) with air temperature and SMAP FT products doesn't rely on the sub-grid open water fraction, tree cover, precipitation, and snowmelt much. The terrain is a major factor for sure because we can see Tibet and Rocky mountains in Figure 9. However, even without considering so many factors, the new algorithm shows comparable agreement with SMAP's FT products. We can not say the new algorithm is better, but that's already inspiring。

In SMAP-FT products, sub-grid open water fraction, terrain heterogeneity, tree cover, precipitation, and snowmelt are not fully discussed yet. The assessment for SMAP FT products is restricted with core site soil temperature and WMO weather station. We are trying to contact the SMAP FT team to get their ground data for considering the landscape, and we believe this will improve the new algorism in the future.

Additional edits are noted below:

Line 66: Are the limitations not clearly described? Authors should include what the limitations are in more details.

Reply: the sentence is revised.

Line 72-74: This is not clear to me. Author should clarify it.

Reply: the sentence is clarified by Eq. 8. But it should not be mentioned in here. So I deleted it from the text.

Line 87: Authors should justify why you used 36km instead of 9km brightness temperature (Tb) data records. Indeed, SMAP data are provided at both 36-km and 9-km spatial resolution. The 9-km spatial resolution is closer to 0.1 degree ERA5.

Reply: As noted on https://nsidc.org/data/SPL3FTP_E/versions/3 "is derived from SMAP enhanced Level-1C brightness temperatures (SPL1CTB_E)." For SPL1CTB_E, Backus-Gilbert optimal interpolation techniques are used to extract enhanced information from SMAP antenna temperatures before they are converted to brightness temperatures. The authors are not familiar with Backus-Gilbert optimal interpolation techniques, and the interpolation will certainly reshape the original DAV signal. Thus, we prefer the raw 36km F/T products.

Line 89: This study used older version of SMAP data.

Reply: We use data downloaded from https://nsidc.org/data/SPL3FTP/versions/3. This is the most recent version of these data. The TB data is stored in the same file as the FT products.

Line 92, 98: ERA5 data provide hourly. What time did authors use for agreement assessment? Is it 6PM or 6AM? Authors should include data source (e.g., web link).

Reply: "According to longitude, the hourly data are interpolated to 6 am and 6 pm local time." is added in Line 97. This is the same method we used in Lv et al. 2016 RSE. The data source is

added in Line 90.

Line 118-123: The relevant citation should be included (Xu?, Derksen? Kim?).

Reply: Xu et al., 2018 is added in Line 128.

Line 134: Surface air temperature from global weather stations were used for landscape FT classification accuracy assessment, not for validation. Authors should check and revise it.

Reply: Thank you very much. "accuracy assessment" replaces "validation" through the text. "reanalysis data" is deleted from Line 133.

Line 165: Why did you use H-pol? Is there any justification?

Reply: "The DAV signals between H and V polarizations have few differences(Sharifnezhad et al., 2021)" is added in Line 175.

Line 178-179: Is this your assumption?

Reply: Thank you. We use "assume" instead of "define" in Line 188.

Line 212: Authors should include in-situ data description in Data sections (e.g., relevant references, data source (web site)).

Reply: There is in-situ data used in this study. At the Xilinhot site, the ERA5 reanalysis data and SMAP data covering the same location are used.

Line 222: Figure 4 does not show soil moisture variations. How did you provide the influence of soil moisture on Tb? If it is soil moisture influence, how much variation in soil moisture?

Reply: The influence of soil moisture on Tb is implicitly included in the TB signal. Figure 3 shows the freezing line as well as the SMAP's FT state, which indicates that the soil is frozen.

Line 263: The geographic location of Xilinhot site should be provided to check if this site is within a domain applied to SCV algorithm in SMAP FT Products.

Reply: Thank you for your suggestion. "(43°30′–45°N, 115°–117°E)" is added in Line 117. The location of Xilinhot is marked in Figure 9, where the comparison is made between the new algorithm and SMAP's FT products. SMAP FT Products downloaded from https://nsidc.org/data/SPL3FTP/versions/3 and applied to the entire north hemisphere.

Line 286: SMAP FT sate products were compared new FT data. Authors should provide more details on SMAP FT state products used in this validation. Which overpass time did you use? (e.g., 6am or 6pm?).

Reply: Section 2.4 explains in detail that both 6 am and 6 pm SMAP products are used and how we construct the daily SMAP FT dataset matching the time resolution of the new algorithm. As suggested by your previous comments, there is no validation in this study but an accuracy assessment.

Line 293: Authors compared two FT state data with different spatial resolution. You should include how to reproject one data from another in method sections. Is it from 0.1 degree to 36km?

Reply: The two FT state data are derived from both SMAP's 36 km data. The SMAP's FT product files contain the TB raw data already. So there is no spatial resolution difference between these two FT state data. "All elements from ERA5-land are interpolated to SMAP 36 km resolution in this study." In Line 107

Line 296: Why was it worse in latitudes above 60N and low latitudes below 30N? Is it false frozen or thawing? What if you use skin or/and soil temperature? Could it be a better agreement?

Reply: The agreement is worse or better does not mean the false frozen or thawing. In SMAP

FT official handbook (Page 24) and Kim et al. 2021 (conclusion part) on Remote Sensing, the agreement between SMAP FT product mission requirement is 80%. According to Kraatz et al. 2018, agreement with 0–5 cm soil temperature at SMAP grids containing CVS stations is about 70%.

Although using the skin/soil temperature may make the results looks better, we would not recommend this in the accuracy assessment because 1) the skin temperature does not correspond with the sensing depth of L-band and 2) the soil temperature from ERA-5-land is criticized in previous studies as "ERA5 is a model reanalysis data with uncertainty as well", so the soil temperature is not reliable for validation. What we did with the ERA5-land is the accuracy assessment by comparing with SMAP FT products and ERA-5 land air temperature.

Air temperature is used in SMAP FT official handbook (Section 4.2.3 CALIBRATION AND VALIDATION in Algorithm Theoretical Basis Document (ATBD) SMAP Level 3 Radiometer Freeze/Thaw Data Products (L3_FT_P and L3_FT_P_E)). Since we lack in-situ soil temperature data as in that handbook, air temperature is the last option we can take in this study.

Line 324: Some studies reported the results on FT accuracy assessment with soil temperature derived FT state. Authors should discuss the results from previous studies.

Reply: "Some studies use soil temperature to evaluate SMAP FT products, and with 0–5 cm, soil temperature at SMAP grids containing CVS stations is about 70%. " is added in Line 341-342. As the core idea in the discussion, we agree that a lack of in situ soil temperature observations presents a key data gap in assessing frozen soil extents.

Line 329-331: Because you did not use soil temperature (indeed, soil temperature from one site only), this statement is not clear conclusion.

Reply: soil temperature is not adopted as explained above. This part does not discuss soil temperature either. The assessment for SMAP FT products is restricted with core site soil temperature and WMO weather station. We are trying to contact the SMAP FT team to get their ground data for considering the landscape

Line338: Is spatial resolution of ERA5 1degree? In data section, the resolution is 0.1 degree.

Reply: Thank you. It is corrected as "0.1°x0.1° lat-lon grid (0 to 100 km$^2$)".

Figure 1: It would be great to include the latitude/longitude of Xilinhot site.

Reply: Thank you. "(43°30′–45°N, 115°–117°E)" is added

Figure 3: It is too complicated. Author could remove unnecessary time-series lines.

Reply: Figure 3 removes soil temperature lines.

Figure 4: Where (or what) is Maqu?

Reply: Thank you. The title part is removed.

Figure 5: Authors should describe study domain in details. E.g., how to define your domain?

Reply: "and we focus on the domain from 20°N to    85.044°N" is added in Line 87 as well as Figure 5.